# A Systematic Review of Intermittent Theta Burst Stimulation for Neurocognitive Dysfunction in Older Adults with Schizophrenia

**DOI:** 10.3390/jpm13030485

**Published:** 2023-03-08

**Authors:** Xinyang Zhang, Xinhu Yang, Zhanming Shi, Rui Xu, Jianqiang Tan, Jianwen Yang, Xiong Huang, Xingbing Huang, Wei Zheng

**Affiliations:** 1The Affiliated Brain Hospital of Guangzhou Medical University, Guangzhou 510260, China; 2Laboratory of Laser Sports Medicine, School of Sports Science, South China Normal University, Guangzhou 510260, China; 3Chongqing Jiangbei Mental Health Center, Chongqing 400000, China

**Keywords:** intermittent theta burst stimulation, schizophrenia, systematic review, older adults, neurocognitive function

## Abstract

Objective: Neurocognitive dysfunction is thought to be one of the core clinical features of schizophrenia, and older adults with schizophrenia exhibited greater overall cognitive deficits than younger adults. The aim of this systematic review was to examine the neurocognitive effects of intermittent theta burst stimulation (iTBS) as an adjunctive treatment for older adults suffering from schizophrenia. Methods: Randomized double-blinded controlled trials (RCTs) investigating the neurocognitive effects of adjunctive active iTBS versus sham iTBS in older adults with schizophrenia were systematically identified by independent investigators searching Chinese and English databases. Results: Two double-blinded RCTs (n = 132) compared the neurocognitive effects of adjunctive active iTBS (n = 66) versus sham iTBS (n = 66) in patients that fulfilled the inclusion criteria of this systematic review and were analyzed. One RCT found significant superiority of active iTBS over sham iTBS in improving neurocognitive performance in older adults with schizophrenia. In the other RCT, the findings on the neurocognitive effects of iTBS as measured by three different measurement tools were inconsistent. The dropout rate was reported in the two RCTs, ranging from 3.8% (3/80) to 7.7% (4/52). Conclusion: There is preliminary evidence that adjunctive iTBS may have some beneficial effects in the treatment of neurocognitive function in older patients with schizophrenia. Future RCTs with larger sample sizes focusing on the neurocognitive effects of adjunctive iTBS in older adults with schizophrenia are warranted to verify these findings.

## 1. Introduction

Neurocognitive dysfunction is thought to be one of the core clinical features of schizophrenia and has been found to be strongly associated with impairments in function. A previous study reported that up to 73% of patients with schizophrenia showed neurocognitive decline [1]. Compared with individuals suffering from mood disorders, neurocognitive impairment in schizophrenia is more severe and tends to be more independent of clinical symptoms [2]. A recent review reported that older adults with schizophrenia exhibited greater overall cognitive deficits than their healthy counterparts and had greater deficits in nearly all cognitive domains than younger adults [3]. Despite advances in psychopharmacologic therapy, the findings for enhancing cognition are mixed and unsatisfactory [4]. Consequently, augmentation strategies of antipsychotic drugs with nonpharmacological interventions, including aerobic exercise [5], cognitive remediation [6] and receptive transcranial magnetic stimulation (rTMS) [7], have been preliminarily used for older adults suffering from schizophrenia in clinical practice.

As a non-invasive physical therapy, rTMS can modulate brain network functioning by producing a local magnetic field that acts on the local cerebral cortex [8]. rTMS has been approved by the Food and Drug Administration of the United States to treat adult patients with major depressive disorder who fail to respond to antidepressants [9]. Numerous randomized controlled trials (RCTs) [10,11] and meta-analyses [12,13] also found the positive treatment effects of rTMS in adult patients suffering from schizophrenia. For example, a recent meta-analysis [12] found that high-frequency rTMS over the left dorsolateral prefrontal cortex (DLPFC) leads to significantly large improvements in treating negative symptoms of schizophrenia. Importantly, accumulating evidence also supports the safety and effectiveness of rTMS in children and adolescents with schizophrenia [14].

As a patterned form of rTMS, theta burst stimulation (TBS), either intermittent (iTBS) or continuous (cTBS) fashion, delivers a shorter stimulation duration and lower stimulation intensity in comparison to conventional rTMS protocols. TBS stimulation can induce more rapid and longer-lasting effects on synaptic plasticity and further regulate functional connectivity in the right posterior parietal cortex [15], which are closely associated with neurocognitive deficits in schizophrenia [16]. As an adjunctive treatment, the beneficial effects associated with TBS were observed in some neurocognitive task domains [17]. A recent review reported the beneficial effects of adjunctive TBS in improving neurocognitive function and auditory hallucinations in subjects with schizophrenia, even in elderly individuals with schizophrenia [18]. Compared with standard rTMS (total duration of 37.5 min), iTBS (total duration of 3 min 9 s) can markedly shorten the treatment duration and achieve rapid-acting antidepressant effects [19]. An animal study found that iTBS rather than cTBS was associated with improvements in learning and strongly reduced cortical protein expression, such as parvalbumin [20]. Furthermore, compared to cTBS, iTBS appeared to have a more favorable effect on the safety of treatments [21,22], which is vital for elderly people. Nevertheless, the research applied to evaluate the potential of adjunctive iTBS for improving neurocognitive function in older adults with schizophrenia has been inconsistent [23,24]. For example, Zhao et al. [23] found that iTBS can improve cognitive function as well as negative symptoms in older adults with schizophrenia. However, another study reported mixed findings on adjunctive iTBS for neurocognitive dysfunction in older adults suffering from schizophrenia [24].

To date, no systematic review on adjunctive iTBS for neurocognitive dysfunction in older adults with schizophrenia has been published. Thus, the aim of the current study was to focus on investigating the neurocognitive effects of adjunctive iTBS in older adults with schizophrenia. Based on the findings of a recent study [25], we hypothesized that adjunctive active iTBS has significant superiority in improving neurocognitive dysfunction for older adults with schizophrenia when compared to sham iTBS.

## 2. Methods

### 2.1. Search Strategy

English (PsycINFO, Cochrane Library, EMBASE, and PubMed) and Chinese (Chinese Journal Net and WanFang) databases were independently searched by two investigators (XYZ and XHY) on 25 January 2022. The search strategy was as follows: (intermittent theta burst stimulation OR iTBS OR intermittent theta burst transcranial magnetic stimulation OR transcranial magnetic intermittent theta-burst stimulation) AND (“schizophrenia”[MeSH] OR schizophrenic disorder OR disorder, schizophrenic OR schizophrenic disorders OR schizophrenias OR dementia praecox) AND (“aged”[MeSH] OR older OR elderly OR senior OR old-age OR aging). Additionally, we manually searched the reference lists from the included RCTs and relevant reviews.

### 2.2. Selection Criteria

In accordance with PRISMA guidelines [26], only studies meeting the following ***PICOS*** criteria were selected. ***P***articipants: Older adults (≥50 years old) suffering from schizophrenia. As recommended, an age ≥50 years has often been used as the cutoff value for “older adults” of the general population [27]. ***I***ntervention versus ***C***omparison: Active iTBS plus antipsychotics versus antipsychotic monotherapy or sham iTBS plus antipsychotics. ***O***utcomes: The primary outcome was the improvement in neurocognitive function as measured by the standardized rating scales such as the Mattis Dementia Rating Scale-2 (MDRS-2). The secondary outcomes recorded in this systematic review were total psychopathology as measured by the Brief Psychiatric Rating Scale (BPRS) [28] and the Positive and Negative Syndrome Scale (PANSS) [29], the dropout rate and adverse events. ***S***tudy: Only published double-blinded RCTs investigating the neurocognitive effects of adjunctive iTBS in older adults suffering from schizophrenia were permitted for inclusion. Observational studies, review articles and case reports/series were excluded.

### 2.3. Data Extraction and Quality Assessment of Each Study

Data from each included RCT were extracted by two investigators (XYZ and XHY). If there were any differences, there was a discussion between the same investigators. A senior author (WZ) was consulted if needed. To acquire missing data from the included RCTs, we contacted the first and/or corresponding authors.

Two investigators (XYZ and XHY) independently assessed the study quality using the Jadad scale [30] and Cochrane risk of bias [31].

## 3. Results

### 3.1. Results of the Search

The PRISMA flow diagram of this systematic review is shown in Figure 1. A total of 100 hits in the English (n = 30) and Chinese databases (n = 70) were identified. Finally, two double-blinded RCTs [23,24] were included in this systematic review.

### 3.2. Sample Characteristics

Table 1 summarizes the characteristics of the two double-blinded RCTs (n = 132) that were included and compares the neurocognitive effects of adjunctive active iTBS (n = 66) versus sham iTBS (n = 66) in older adults with schizophrenia. The two RCTs were conducted in China. The total number of participants included in the analysis was 125 (Table 1). In the included RCTs, the mean age and illness duration were 63.4 years and 17.4 years, respectively. All patients were elderly people (100%) and inpatients (100%). Sessions were conducted five times per week with iTBS for four weeks in the two included RCTs. Male patients accounted for 53.6% of the sample. The use of antipsychotics is mentioned in detail in only one RCT (50%, 1/2) [24], involving a variety of antipsychotics (Table 1), and the doses of antipsychotics used in the two RCTs were 511 mg [23] and 535 mg [24] per day after the conversion to chlorpromazine equivalents, respectively.

### 3.3. Assessment of Study Quality

As shown in Figure 2, two RCTs were assessed as low risk regarding attrition bias and reporting bias according to the Cochrane risk of bias. Only one RCT was rated as low risk with regard to random sequence generation. The Jadad scores ranged from 4 to 5 across the two RCTs, indicating high-quality studies.

### 3.4. Neurocognitive Function

The neurocognitive effects of active iTBS compared with sham iTBS in older adults with schizophrenia are summarized in Table 2. Zhao et al. [23] found significant superiority of active iTBS over sham iTBS in improving neurocognitive function measured by MDRS-2 and its subscales (*p* < 0.05). In the other RCT [24], the findings on the neurocognitive effects of active iTBS compared with sham iTBS in older adults with schizophrenia were inconsistent (Table 2).

### 3.5. Total Psychopathology

As shown in Table 3, only one RCT reported the efficacy of active iTBS versus sham iTBS in treating total psychopathology in schizophrenia [23]. When compared to sham iTBS, active iTBS was significantly associated with improvement in total psychopathology as measured by the PANSS (*p* < 0.05) but not the BPRS (*p* > 0.05).

### 3.6. Dropout Rate and Adverse Events

The dropout rate and adverse events are summarized in Table 4. The dropout rate was reported in the two RCTs, ranging from 3.8% (3/80) to 7.7% (4/52). Only one study reported adverse events, finding no significant group difference.

## 4. Discussion

This study is the first systematic review to examine the neurocognitive effects of adjunctive iTBS in older adults suffering from schizophrenia. Only two RCTs [23,24] were included, involving 132 older adults with schizophrenia. The main findings of this systematic review are the following: first, two RCTs investigated the effects of adjunctive iTBS in treating neurocognitive function in older adults with schizophrenia, suggesting that iTBS may have some beneficial effects; second, only one RCT reported the efficacy of active iTBS versus sham iTBS in treating total psychopathology as measured by the PANSS and BPRS in schizophrenia, with mixed findings; and third, iTBS appeared to be safe and tolerable as an adjunct treatment in older adults with schizophrenia.

The main purpose of investigating iTBS is to monitor its neurocognitive effects, which has been achieved via multiple neurotherapeutic strategies including magnetic seizure therapy (MST), transcranial direct current stimulation (tDCS) and nonconvulsive electrotherapy (NET) [32]. Several studies have found that NET appears to be effective as an adjunct treatment in schizophrenia [32] and treatment-refractory unipolar and bipolar depression [33,34,35] without serious associated neurocognitive impairments. However, the findings of the two RCTs [23,24] included in this systematic review on the neurocognitive effects of adjunctive iTBS for older adults with schizophrenia were inconsistent. Specifically, Zhao et al. [23] found iTBS affected all neurocognitive modalities, including five subscales (attention, initiation/perseveration, conceptualization, construction and memory), which reflected an overall improvement in attentional function, executive function, visuospatial skills and memory function [36]. However, Zhen et al. [24] found different effects of iTBS on neurocognitive patterns. Similarly, a recent systematic review reported inconsistent findings on the neurocognitive effects of MST in schizophrenia [37]. For example, Jiang et al. found that MST showed significantly less cognitive impairment with regard to immediate memory, language function, delayed memory and global cognitive function when compared to ECT [38]. Another study reported that MST led to a specific deficit in the Autobiographical Memory Inventory-Short Form (AMI-SF) without affecting other cognitive domains [39]. Taken together, the neurocognitive effects of neurotherapeutic strategies including iTBS and MST in patients with schizophrenia should be further investigated.

The mechanism of iTBS in improving the neurocognitive function of schizophrenia is unclear. A possible explanation for the mechanism is that iTBS could alter neuronal excitability levels in the primary motor cortex, which may affect neurocognitive function [40]. Although a meta-analysis of cognitive function may not be possible due to differences in measurements, we still found preliminary evidence that iTBS may have a modest enhancing effect on cognitive function in older adults with schizophrenia. Future studies focusing on the neurocognitive effects of adjunctive iTBS in older adults with schizophrenia need to be performed with specific cognitive batteries, such as the MATRICS (Measurement and Treatment Research to Improve Cognition in Schizophrenia) [41].

In this systematic review, only one RCT examined the effects of iTBS on total psychopathology as measured by the PANSS and BPRS in older adults with schizophrenia, and there were mixed findings [23]. Accumulating evidence has demonstrated the positive effects of iTBS on total psychopathology in younger adults with schizophrenia [25,42,43]. For example, a recent RCT found that iTBS had a particularly prominent effect as an adjunct treatment in improving the total symptoms in younger adults with schizophrenia [43]. In general, older adults suffering from schizophrenia have been associated with higher levels of depressive and anxiety symptoms than younger adults [44,45], which predicted poorer treatment outcomes. However, a recent study did not support the age-related differences in the antidepressant effects of citalopram in schizophrenia patients with subsyndromal depression [46]. The comparison of efficacy and safety of adjunctive iTBS in older versus younger adults with schizophrenia has not been investigated.

In this systematic review, the observed rates of discontinuation for any reason and adverse events were low in the iTBS groups, suggesting the good acceptability and the feasibility of adjunctive iTBS for older patients suffering from schizophrenia in clinical practice. Similarly, one systematic review focusing on the safety of adjunctive TBS for the general population found that 5% of subjects reported mild adverse events [47]. Another systematic review including studies focusing on children and adolescent patients found that TBS interventions were associated with mild adverse events [48]. Similarly, other neurotherapeutic strategies such as tDCS and tACS (transcranial alternating current stimulation) were safe and effective in treating patients with schizophrenia. For example, tDCS was effective and safe in ameliorating negative symptoms in patients with schizophrenia [49]. A recent review revealed that tACS was also a safe and effective strategy in improving neurocognitive function of patients with schizophrenia [50]. However, there have been no head-to-head studies comparing the safety of iTBS either with tDCS or tACS for older patients with schizophrenia.

The following limitations of this systematic review should be considered. First, given the heterogeneity in significance between the included RCTs, and the different methods they employed (for example, neurocognitive function was measured by four different assessment tools in the two RCTs), a meta-analysis could not be performed. Second, the small sample sizes (n = 132), ranging from 52 to 80, potentially reduced the studies’ power and increased the possibility of type II error. Third, only one RCT reported adverse events when evaluating the clinical efficacy of iTBS in older adults with schizophrenia. Finally, this systematic review on adjunctive iTBS for older adults suffering from schizophrenia has not been registered.

## 5. Conclusions

There is preliminary evidence for the beneficial effects of adjunctive iTBS in treating neurocognitive function in older adults with schizophrenia. Future RCTs with larger sample sizes focusing on the neurocognitive effects of adjunctive iTBS in older adults with schizophrenia are warranted to verify these findings.

## Figures and Tables

**Figure 1 jpm-13-00485-f001:**
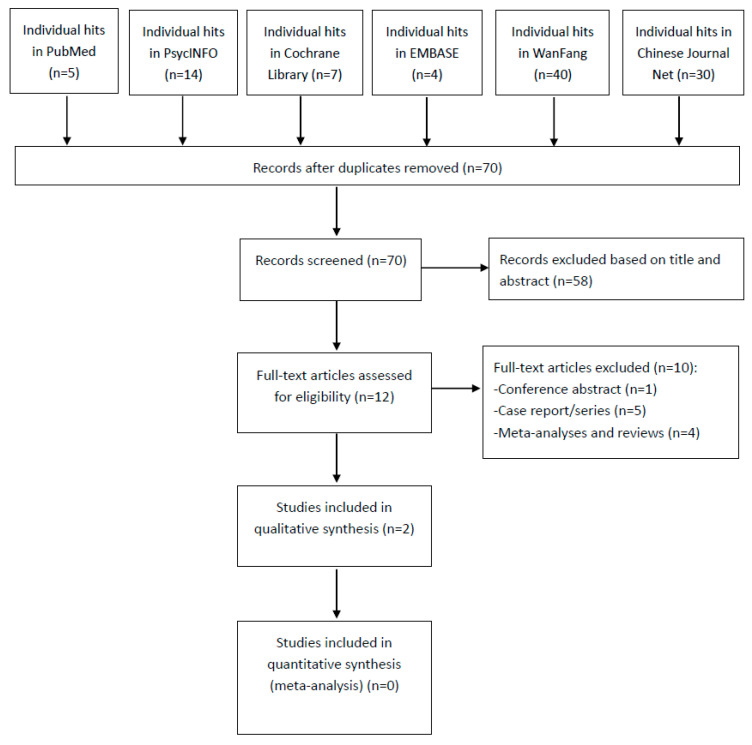
PRISMA flow diagram.

**Figure 2 jpm-13-00485-f002:**
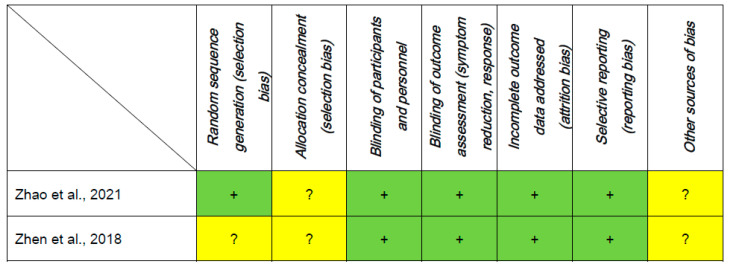
Risk of bias [23,24]. Abbreviations: + or green: low risk of bias; ? or yellow: unclear risk of bias.

**Table 1 jpm-13-00485-t001:** Summary of the characteristics of the included studies.

Study (Country)	N ^a^	Sex ^b^: Male (%)	Diagnosis (%)	Diagnostic Criteria	Age ^b^: yrs (Range)	Duration ^b^ of Illness (yrs)	-Design -Analysis -Setting	AP: Dose (mg/d)	Intervention: -Device -Brain region -Stimulation Intensity	Treatment Duration (n/wks)	Jadad Scores
Zhao et al., 2021 (China)	total: 52 iTBS: 26 sham: 26	66.7	SCZ (100%)	ICD-10	63.3 (59–71)	33.2	-DB -OC -inpatients	iTBS: CPZ-equ = 495 sham: CPZ-equ = 527	-NR -L-DLPFC -iTBS (100% MT) versus sham (100% MT, tilt the coil 90 degrees from the original stimulation position relative to the scalp)	4 (5/wks)	5
Zhen et al., 2018 (China)	total: 80 iTBS: 40 sham: 40	45.5	SCZ (100%)	DSM-IV	63.4 (NR)	7.5	-DB -OC -inpatients	iTBS: CPZ-equ = 554 ^c^ sham: CPZ-equ = 516 ^c^	-Magpro R100 (Tonica Elektronik A/S) -L-DLPFC -iTBS (80% MT) versus sham (80% MT, tilt the coil 180 degrees from the original stimulation position relative to the scalp)	4 (5/wks)	4

^a^ Data were extracted based on random assignment and the total number of participants included in the analysis was 125, of which 48 were from Zhao et al. (2021) and 77 were from Zhen et al. (2018). ^b^ Available data were extracted based on the mean baseline value of each included trial. ^c^ Including risperidone (n = 24), olanzapine (n = 22), aripiprazole (n = 9), quetiapine (n = 17), chlorpromazine (n = 4), ziprasidone (n = 4). Abbreviations: AP = antipsychotics; CPZ-equ = chlorpromazine equivalents; DB = double blind; DSM-IV = Diagnostic and Statistical Manual of Mental Disorders, 4th version; ICD-10 = International Classification of Diseases, 10th version; iTBS = intermittent theta burst stimulation; L-DLPFC = left dorsolateral prefrontal cortex; MT = motor threshold; n = number of times; N = number of patients; NR = not reported; OC = observed case; SCZ = schizophrenia; sham = sham stimulation; wks = weeks; yrs = years.

**Table 2 jpm-13-00485-t002:** Improvement in cognitive function after iTBS.

Study	Cognitive Function	Pre-iTBS (Mean ± SD)	Post-iTBS (Mean ± SD)	Pre-Sham (Mean ± SD)	Post-Sham (Mean ± SD)	Findings ^a^
Zhao et al., 2021	**MDRS-2:**					
	Total scores	109.7 ± 6.8	116.5 ± 6.7	107.9 ± 7.5	108.7 ± 7.3	** *p* ** ** < 0.05**
	Attention	30.3 ± 2.0	32.1 ± 2.0	30.3 ± 2.2	30.4 ± 2.2	** *p* ** ** < 0.05**
	Initiation/perseveration	31.1 ± 1.8	33.3 ± 1.8	30.3 ± 2.2	30.5 ± 2.2	** *p* ** ** < 0.05**
	Conceptualization	30.0 ± 1.5	31.4 ± 2.0	30.0 ± 1.9	30.3 ± 1.8	** *p* ** ** < 0.05**
	Construction	4.9 ± 0.7	5.3 ± 0.5	4.6 ± 0.6	4.7 ± 0.6	** *p* ** ** < 0.05**
	Memory	13.3 ± 2.1	14.4 ± 2.5	12.7 ± 1.6	12.9 ± 1.6	** *p* ** ** < 0.05**
Zhen et al., 2018	**Digit span test:**					
	Total scores	15.6 ± 4.1	16.1 ± 4.2	16.2 ± 4.4	15.8 ± 3.9	NS
	Digit span forward	10.6 ± 2.5	11.3 ± 1.9	10.8 ± 2.1	10.6 ± 2.5	** *p* ** ** < 0.05**
	Digit span backward	5.2 ± 2.5	5.4 ± 2.1	5.1 ± 1.9	4.9 ± 1.6	NS
	**Spatial span test:**					
	Total scores	12.0 ± 4.4	16.3 ± 4.2	11.9 ± 4.1	10.2 ± 3.9	** *p* ** ** < 0.05**
	Spatial span forward	5.8 ± 2.1	7.5 ± 2.2	5.6 ± 1.9	6.2 ± 2.1	** *p* ** ** < 0.05**
	Spatial span backward	6.2 ± 2.4	6.9 ± 3.1	6.5 ± 1.7	5.9 ± 2.4	NS
	**WCST:**					
	Response administered	118.0 ± 14.0	120.0 ± 17.0	121.0 ± 15.0	120.0 ± 11.0	NS
	Categories completed	4.7 ± 1.3	5.2 ± 1.1	4.7 ± 1.1	4.6 ± 1.3	NS
	Total correct responses	50.0 ± 10.0	54.0 ± 9.0	51.0 ± 9.0	49.0 ± 10.0	NS
	Total errors	70.0 ± 18.0	63.0 ± 22.0	69.0 ± 21.0	72.0 ± 19.0	NS
	Percent errors	59.8 ± 14.7	51.5 ± 12.2	56.4 ± 14.1	58.2 ± 12.1	** *p* ** ** < 0.05**
	Trials to complete first category	25.2 ± 13.5	21.7 ± 14.1	23.5 ± 9.7	25.0 ± 11.4	NS
	Perseverative responses	37.0 ± 11.5	41.1 ± 12.3	35.9 ± 13.2	39.8 ± 14.3	NS
	Perseverative errors	55.3 ± 18.0	44.2 ± 17.6	51.5 ± 20.1	56.8 ± 22.3	** *p* ** ** < 0.05**
	Percent perseverative errors	71.3 ± 12.6	62.7 ± 11.4	66.1 ± 11.7	70.5 ± 14.5	** *p* ** ** < 0.05**
	Nonperseverative errors	17.4 ± 5.1	17.1 ± 7.5	18.6 ± 6.7	20.9 ± 5.8	NS
	Percent nonperseverative errors	28.7 ± 12.2	30.3 ± 13.1	29.7 ± 11.4	32.1 ± 13.6	NS
	Failure to maintain set	14.9 ± 6.8	13.3 ± 7.6	11.5 ± 6.7	13.5 ± 5.4	NS
	Percent conceptual level responses	53.3 ± 17.0	63.5 ± 15.9	51.8 ± 18.8	56.9 ± 17.6	** *p* ** ** < 0.05**

Abbreviations: iTBS = intermittent theta burst stimulation; MDRS-2 = Mattis Dementia Rating Scale-2; NS = not significant; WCST = Wisconsin Card Sorting Test. ^a^ Reflect the differences between groups in change between pre- and post-intervention values.

**Table 3 jpm-13-00485-t003:** Improvement in clinical symptoms after iTBS.

Study	Clinical Effects	Pre-iTBS (Mean ± SD)	Post-iTBS (Mean ± SD)	Pre-Sham (Mean ± SD)	Post-Sham (Mean ± SD)	Findings ^a^
Zhao et al., 2021	**BPRS**	32.1 ± 1.9	31.3 ±1.9	32.3 ± 2.4	32.2 ± 2.3	NS
	**PANSS**	60.2 ± 3.7	56.7 ± 3.1	61.0 ± 2.8	60.7 ± 2.7	** *p* ** ** < 0.05**
Zhen et al., 2018	NR					

Abbreviations: BPRS = Brief Psychiatric Rating Scale; iTBS = intermittent theta burst stimulation; NR = not reported; NS = not significant; PANSS = Positive and Negative Syndrome Scale. a Differences between groups in change between pre- and post-intervention values. ^a^ Differences between groups in change between pre- and post-intervention values.

**Table 4 jpm-13-00485-t004:** Rates of dropout and adverse events.

Study	Adverse Events				Dropout Rate		
	Events	Total (%)	iTBS Group (%)	Sham Group (%)	Total (%)	iTBS Group (%)	Sham Group (%)
Zhao et al., 2021	Intolerance	1 (1.9)	1 (3.8)	0	4 (7.7)	2 (7.7)	2 (7.7)
Zhen et al., 2018	NR				3 (3.8)	2 (5.0)	1 (2.5)

Abbreviations: AEs = adverse events; iTBS = intermittent theta burst stimulation; NR = not reported.

## Data Availability

The data that support the findings of this study are available from the corresponding author on reasonable request.

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
