# Peer review of "A Systematic Review of Intermittent Theta Burst Stimulation for Neurocognitive Dysfunction in Older Adults with Schizophrenia"

_jpm, 2023, doi:10.3390/jpm13030485_

Round 1

Reviewer 1 Report

A systematic review of intermittent theta burst stimulation for neurocognitive dysfunction in older adults with schizophrenia

This systematic review examines the neurocognitive effects of intermittent theta burst stimulation (iTBS) as an adjunctive treatment for older adults suffering from schizophrenia. I found the manuscript well written and summarizes the findings despite the scarcity of data.

However, the findings of the two RCTs included in the present manuscript on the neurocognitive effects of adjunctive iTBS in older adults with schizophrenia were inconsistent. So, this conclusion of the beneficial effect of iTBS is drawn from a single study and may need more research.

But the present manuscript may help advance the understanding of adjunctive iTBS in older adults with schizophrenia.

Author Response

 We agree with this comment, and we have made it correction as follows:

“There is preliminary evidence for the beneficial effects of adjunctive iTBS in treating neurocognitive function in older adults with schizophrenia. More future RCTs with larger sample sizes focusing on the neurocognitive effects of adjunctive iTBS in older adults with schizophrenia are warranted to verify these findings.”  (page 16, line 1, Conclusion)

Reviewer 2 Report

This report on theta burst stimulation in elderly SCZ patients is fairly written. Some suggestions from a reader's prescpective would be that the reader would like to know what was the primary treatment that the patiets were on if TBS was used as an "adjunctive treatment". There is no mention anywhere in the manuscript what the primary treatment pt's were on. 

As the authors discuss in their limitations of the manuscript, really 2 RCT's are not enough (even though it is highlighted that there were 132 total number of pts in both studies combined) to reach any conclusions at all, particularly when one of the RCT's had inconclusive results. Another useful addition to the manuscript is to detail what the differences and similarities in both these RCT’s are and how they differ in their conclusions.

Another change that would make the manuscript acceptable to the readers is to discuss in depth what these neurocognitive modalities that TBS improves. There is fleeting mention of these neurocognitive modalities in tabular form but no description anywhere in the manuscript. Does TBS affect all modalities similarly or differently? What are the differences and similarities seen in neurocognitive modalities that improved with TBS in pts in at least one of the RCT’s ?

Author Response

We thank the Reviewers and the Editor for the helpful comments on our manuscript. We have revised the manuscript according to the Reviewers’ and Editor’s comments and recommendations. In the revised manuscript, all the changes are in yellow colour for easy inspection.

1.This report on theta burst stimulation in elderly SCZ patients is fairly written. Some suggestions from a reader's prescpective would be that the reader would like to know what was the primary treatment that the patients were on if TBS was used as an "adjunctive treatment". There is no mention anywhere in the manuscript what the primary treatment pt's were on. 

Authors’ reply: As shown in Table 1, TBS was used as an "adjunctive treatment" in the two included RCTs. Furthermore, we have made it clear on the primary treatment in the revised manuscript. The results are as follows:

“The use of antipsychotics is mentioned in detail in only one RCT (50%, 1/2) [24], involving a variety of antipsychotics (Table 1), and the doses of antipsychotics used in the two RCTs were 511 mg [23] and 535 mg [24] per day after the conversion to chlorpromazine equivalents, respectively.” (page 8, line 1)

 2.As the authors discuss in their limitations of the manuscript, really 2 RCT's are not enough (even though it is highlighted that there were 132 total number of pts in both studies combined) to reach any conclusions at all, particularly when one of the RCT's had inconclusive results. Another useful addition to the manuscript is to detail what the differences and similarities in both these RCT’s are and how they differ in their conclusions.

Authors’ reply: We accept these valuable comments and this point has been addressed in the revised results and discussion section as follows:

“All patients were elderly people (100%) and inpatients (100%). Sessions were conducted five times per week with iTBS for four weeks in the included two RCTs.” (page 8, line 2, result section)

And

“Specifically, Zhao et al. [23] found iTBS affected all neurocognitive modalities, including five subscales- attention, initiation/perseveration, conceptualization, construction and memory, which reflected an overall improvement in attentional function, executive function, visuospatial skills and memory function [36]. However, Zhen et al. [24] found different effects of iTBS on neurocognitive patterns. “(page 13, line 15, discussion section)

3. Another change that would make the manuscript acceptable to the readers is to discuss in depth what these neurocognitive modalities that TBS improves. There is fleeting mention of these neurocognitive modalities in tabular form but no description anywhere in the manuscript. Does TBS affect all modalities similarly or differently? What are the differences and similarities seen in neurocognitive modalities that improved with TBS in pts in at least one of the RCT’s ?

Authors’ reply: We accept these valuable comments and this point has been revised in the revised discussion section as follows:

“Specifically, Zhao et al. [23] found iTBS affected all neurocognitive modalities, including five subscales- attention, initiation/perseveration, conceptualization, construction and memory, which reflected an overall improvement in attentional function, executive function, visuospatial skills and memory function [36]. However, Zhen et al. [24] found different effects of iTBS on neurocognitive patterns. Similarly, a recent systematic review reported inconsistent findings on the neurocognitive effects of MST in schizophrenia [37]. For example, Jiang et al. found that MST showed significantly less cognitive impairment with regard to immediate memory, language function, delayed memory, and global cognitive function when compared to ECT [38]. Another study reported that MST would lead to a specific deficit in the Autobiographical Memory Inventory Short Form (AMI-SF) without affecting other cognitive domains [39]. Taken together, the neurocognitive effects of neurotherapeutic strategies including iTBS and MST in patients with schizophrenia should be further investigated.

The mechanism of iTBS in improving the neurocognitive function of schizophrenia is unclear. Possible explanation for the mechanism is that iTBS could alter neuronal excitability levels in the primary motor cortex, which may affect neurocognitive function [40]. Although a meta-analysis of cognitive function may not be possible due to differences in measurements, we still found preliminary evidence that iTBS may have a modest enhancing effect on cognitive function in older adults with schizophrenia. Future studies focusing on the neurocognitive effects of adjunctive iTBS in older adults with schizophrenia need to be performed with specific cognitive batteries, such as the MATRICS (Measurement and Treatment Research to Improve Cognition in Schizophrenia) [41]. ” (page 13, line 15, discussion section).